# Detecting Fraudulent Financial Reporting Using the Fraud Hexagon Model: Evidence from the Banking Sector in Indonesia

**Tarmizi Achmad [1], Imam Ghozali [1], Monica Rahardian Ary Helmina [2], Dian Indriana Hapsari [3] and Imang Dapit Pamungkas [3,*]**

1   Faculty of Economics and Business, Universitas Diponegoro, Semarang 50275, Indonesia
2   Faculty of Economics and Business, Universitas Lambung Mangkurat, Banjarmasin 70123, Indonesia
3   Faculty of Economics and Business, Universitas Dian Nuswantoro, Semarang 50131, Indonesia
*   Correspondence: imangdapit.pamungkas@dsn.dinus.ac.id

**Abstract:** The purpose of this study was to examine the potential for fraudulent financial reporting using the fraud hexagon theory factors such as stimulus (financial target, financial stability, and external pressure), capability (change in director), collusion (total board of commissioners who have multiple positions), opportunity (ineffective monitoring), rationalization (auditor switching), and arrogance (frequency of the number of photos of the chief executive officer (CEO) in the annual financial statements) affect fraudulent financial reporting. The sample of this study comprises banking companies listed on the Indonesia Stock Exchange (IDX) in 2017–2021, with a total sample of 215 and data processing using SPSS 25 software. The results of this study indicate that external pressure and arrogance affect fraudulent financial reporting. However, financial targets, financial stability, ineffective monitoring, auditor switching, change in director, and collusion do not affect fraudulent financial reporting. Therefore, for a company to have a system for preventing the occurrence of embezzlement, the company has to create a system of detection, monitoring, and systems review policies in the field of human resources (HR).

**Keywords:** fraudulent financial reporting; fraud hexagon; collusion

**JEL Classification:** G32; M14; M48

## 1. Introduction

The Association of Certified Fraud Examiners (ACFE), with updated data processing in 2019, obtained the results that corruption is the most detrimental type of fraud in Indonesia. Based on the survey results presented by the ACFE, 69.9% of 167 respondents stated that corruption is the most dangerous type of fraud in Indonesia. The second place is occupied by the misuse of state and company assets/wealth, selected by 20.9% or 50 respondents. Finally, 9.2% or 22 respondents opted for fraudulent financial reporting, a minor proportion. The impact of losses occurs between Indonesian rupiah (IDR) 100 million and IDR 500 million per case, where corruption is detected in a reasonably short period (less than 12 months). A survey conducted by the ACFE in 2019 shows that corruption accounts for most of the losses above IDR 10 million. However, the most common fraud is caused by fraudulent financial reporting.

In the 2019 fraud survey in Indonesia, there were 239 cases of fraud, with details of 167 cases of corruption, 50 cases of misuse of state and company assets/wealth, and 22 cases of fraudulent financial reporting. The total loss caused by fraud reached IDR 873,430,000,000. Of this total, corruption accounted for 69.9%, representing a total loss of IDR 373,650,000,000; misappropriation of state and company assets/wealth was 20.9%, for a total loss of IDR 257,520,000,000; and fraudulent financial reporting was 9.2% with a

total loss of IDR 242,260,000,000. The average loss caused per case was IDR 7,248,879,668. Although the total number of claims and losses caused by fraudulent financial reporting is the smallest, based on the survey results, fraudulent financial reporting is the most common fraud act.

According to information compiled by Consumer News and Business Channel Indonesia (CNBC) Indonesia, the case of credit card manipulation at Bukopin has been going on for more than 5 years, with more than 100,000 credit cards being modified. This discrepancy was recognized by the internal management of Bank Bukopin and reported to a public accounting firm, after which direction decided to restate the figure. The case arose in 2018, where initially, the administration proposed restating or resubmitting the 2017 financial statements. This was due to corrections in the 2015 and 2016 financial statements, which contained misstatements in credit receivables deemed unreasonable on bank credit cards and other balances allowance for impairment losses on assets. Carried over the error from the internal audit of Bank Bukopin, Kantor Akuntan Publik (KAP), which acts as an independent auditor, Bank Indonesia, which has a role as the payment system authority, and Otoritas Jasa keuangan (OJK) as a banking supervisory institution. After the findings, OJK asked for clarification from Bukopin and the independent audit agency, which at that time was the Purwantono, Sungkoro, and Surja KAP, where the KAP was an affiliate of KAP Ernst & Young. This case has two possibilities, namely, a misstatement or fraudulent misstatement. The first allegation of the statement that has been found is related to factors that have the potential for fraud, specifically the ineffective monitoring of the relevant supervisors, namely the internal audit of Bank Bukopin, the related KAP, and OJK (CNBCIndonesia 2018).

Many cases of banking sector fraud have occurred in Indonesia. In 2018 Bank Pembangunan Daerah Jawa Barat dan Banten (BJB) Syariah was one of the companies that suffered losses due to fraud. The issue surrounding Bank BJB Syariah is an alleged fictitious credit with a loss value of IDR 548 billion. Based on the 2018 good corporate governance (GCG) report, the company noted four cases of internal fraud that significantly affected the bank's operational activities and financial condition in 2018. The strong suspicion underlying the fraud was due to the opportunity for permanent employees of Bank BJB Syariah to manipulate data. The manipulation is carried out when customers want to apply for credit, but the employees consciously manipulate the amount of credit submitted (FinancialBussiness.com 2019). One of the banking companies owned by state-owned enterprises, namely BNI—more precisely, the Bank Negara Indonesia (BNI) Ambon Branch Office—found a case of theft of customer deposit funds in 2019. The loss due to the theft reached a value of IDR 134 billion. In collaboration with employees of the Bank BNI Ambon branch office, employees of the Bank BNI Makassar branch office carried out the theft. There were unusual transactions and investments in the accounts of the two suspects. The report on the alleged case was discovered because of suspicions from other BNI Ambon branch office employees of a procedural violation. The results of the internal audit revealed that the violation did indeed occur, which was followed by an investigation of the case.

In 2020, Maybank became a hot topic of discussion regarding the loss of customer funds amounting to IDR 22.9 billion. The funds were savings belonging to a Maybank customer at the Cipulir branch office. The loss of customer funds is suspected to be due to a theft that led to fraud. The suspect in the case is the head of the Maybank Cipulir Branch Office. The funds were used for personal purposes. It is challenging to minimize losses while attempting to change the economic structure (Batrancea et al. 2021, 2022). The fraud was allegedly related to the abuse of authority by the office owner (CNBCIndonesia 2020).

The novelty of this study is to test the fraud hexagon. The latest fraud model with eight factors adds a collusion variable. Using the total number of commissioners with concurrent positions to analyze ways to prevent fraudulent financial reporting in the banking sector in Indonesia. This study examines the potential of fraudulent financial reporting using the fraud hexagon theory based on factors such as stimulus (financial target, financial stability, and external pressure) and capability (change of directors). Next,

collusion (number of board of commissioners who have multiple positions), opportunity (unmonitored effective), rationalization (auditor switching), and arrogance (frequency of the number of Chief Executive Officer (CEO) photos in the annual financial report) affect fraudulent financial reporting in the banking sector in Indonesia.

This study is divided into five main sections. The first is an introduction which contains the case in the banking sector in Indonesia and the inconsistency of research results on fraudulent financial reporting. Section 2 presents a literature review explaining the eight factors of fraud hexagons on fraudulent financial reporting. This section also includes hypothesis development. In Section 3, the methodology covers variable operational definitions and research sample criteria. Section 4 provides the results, and in Section 5, the discussion and conclusions of the study are explained.

## 2. Literature Review

### 2.1. Effect of Financial Targets in Detecting Fraudulent Financial Reporting

Every company has financial goals to be achieved. Financial targets are profit goals that the company must meet. Return on assets (ROA) is a method of calculating company profits based on company performance. The board of directors or management sets financial and sales targets and profit levels. Indirectly, these financial targets put additional pressure on management to meet the goals that have been set (Sari Pramono et al. 2020).

Hexagon Theory deals with financial targets. According to the hexagon theory, where managers are obliged to seek the maximum profit, they may need to work on carrying out their responsibilities for certain things so that the intended goals are not achieved. Managers will be influenced to commit fraudulent financial reporting by demands to meet financial targets in return for significant incentives. This shows the company's existence because the more significant its ability to meet its financial goals, the better its performance will be (Evana et al. 2019; Pamungkas and Utomo 2018).

These assumptions indicate that financial targets can influence managers to engage in fraudulent financial reporting. Therefore, based on this concept, financial targets are hypothesized to influence fraudulent financial reporting. This statement is supported by previous research by (Nanda et al. 2019; Sari et al. 2020; Sari Pramono et al. 2020; Sawaka and Hiwa 2020; Utami and Pusparini 2019).

**Hypothesis 1 (H1).** *Financial Target affects Fraudulent Financial Reporting.*

### 2.2. Effect of Financial Stability in Detecting Fraudulent Financial Reporting

Financial stability is when the company's financial situation is stable. According to the fraud hexagon theory, managers are under pressure to commit fraudulent financial reporting when the state of the economy, industry, or operational entity threatens financial stability. Management is often under pressure to show that the company has managed assets well enough to generate large profits, ultimately generating high returns for investors (Akbar 2017).

Overall assets managed by the company usually show financial stability from year to year. Companies that have large total assets will be able to provide high profits to investors. On the other hand, if the company's total assets decrease, it can cause investors, creditors, and decision-makers to lose interest because the company's status is unstable and not running properly or profitably. For company managers, a reduction in total or low total assets will put them under much stress. For management to limit the flow of investment capital in the coming year, the financial statements were falsified by management to hide the company's unfavourable and detrimental stability (Irwandi et al. 2019). The assumption concludes that financial stability can cause managers to manipulate financial statements. Based on this theory and the research results of Apriliana and Agustina (2017), Nanda et al. (2019), Situngkir and Triyanto (2020), and Utami and Pusparini (2019), it is hypothesized that financial stability affects fraudulent financial reporting.

**Hypothesis 2 (H2).** *Financial stability affects Fraudulent Financial Reporting.*

*2.3. Effect of External Pressure on Detecting Fraudulent Financial Reporting*

The pressure that the company receives from outside is known as external pressure. External pressure occurs when a business has difficulty paying its high-risk credit debt. The greater the credit risk, the more hesitant lenders lend money to the business. In competition with other companies, they use more funds in the form of additional investment for financing related to the company's operational activities (Manurung and Hardika 2015). As a result, management is under pressure to keep the company's performance competitive. In addition, pressure on management will force them to report financial figures as accurately as possible to assure external parties that the company can repay the loans it has taken.

External pressure is related to the hexagon theory, which states that obtaining additional funds from third parties is one of the pressures that must be faced. Pressure on management will motivate them to do everything they can to continue to present strong financial reports, including falsifying financial records to show good performance to meet outsiders' expectations (Husmawati et al. 2017; Shi et al. 2017). The leverage ratio is used to analyze the company's ability to repay loans to external parties. The ratio of total liabilities to total assets is known as the leverage ratio. If a company has much debt, the credit risk is also higher. The higher the credit risk, the more creditors hesitate to provide loans to the company (Nurcahyono et al. 2021). Therefore, external pressures can influence managers to commit fraudulent financial reporting, as has been assumed. Based on this assumption and supported by previous research, Achmad et al. (2022b), Pamungkas and Utomo (2018), Situngkir and Triyanto (2020), Utami and Pusparini (2019) hypothesized that external pressures affect fraudulent financial reporting.

**Hypothesis 3 (H3).** *External pressure affects Fraudulent Financial Reporting.*

*2.4. Effect of Ineffective Monitoring in Detecting Fraudulent Financial Reporting*

Ineffective monitoring is when a company's performance is monitored through an ineffective system. Ineffective company supervision will encourage managers to commit fraudulent behaviour. Effective surveillance measures can reduce fraudulent practices (Evana et al. 2019; Lou and Wang 2011). Ineffective monitoring occurs when the company's audit committee mechanism needs to be fixed so that monitoring is ineffective. The rise of accounting scandals and fraudulent activities is one consequence of the lack of corporate oversight, allowing individuals to misbehave in their best interests (Manurung and Hardika 2015). Lack of oversight from within the business allows managers to maximize their profits. As a result, third parties, such as independent commissioners, must exercise management oversight to prevent fraud. The board of commissioners is in charge of overseeing the running of the company and providing advice to the board of directors. Company supervision is expected to be more effective with an independent board of commissioners, and fraudulent practices will be reduced. This is so that by selecting commissioners with no relationship with shareholders, directors, management, or other internal parties, the board of commissioners can carry out more balanced supervision (Anggilia and Rinaldo 2015).

Ineffective supervision is related to the hexagon theory, which states that the principal delegates authority to the agent to carry out the principal's goals, but the agent prioritizes his interests when managing the company. Because of this conflict of interest, the principal must supervise the agent because if the company's supervision is ineffective, the agent can commit fraud. Furthermore, the function of an independent commissioner helps reduce the knowledge asymmetry between the principal and the agent, ensuring that the principal's interests are maintained (Manurung and Hardika 2015). Based on these assumptions and in line with the research results of (Husmawati et al. 2017; Kamal et al. 2016; Sinarti 2019; Situngkir and Triyanto 2020), ineffective monitoring affect fraudulent financial reporting, it is concluded that the lack of supervision can result in fraudulent financial reporting.

**Hypothesis 4 (H4).** *Ineffective monitoring affects Fraudulent Financial Reporting.*

*2.5. Effect of Auditor Switching Affects Fraudulent Financial Reporting*

Change in auditor or auditor switching is a practice where companies change auditors. Government-mandated mandatory audit rotations or voluntary modifications may result in auditor changes. The Indonesian government has enacted regulations governing the limitation of offering public accountant audit services to clients (Utomo et al. 2019). Regulation of the Minister of Finance of the Republic of Indonesia Number 17/PMK.01/2008, which regulates public accounting services, regulates the provisions of auditor switching. These changes include a public accounting firm that provides general audit services to the same customer for six consecutive financial years and a public accountant who provides general audit services to the same client for three consecutive financial years. In the context of the hexagon theory, where this theory describes the company's existence, rotation is a must. This theory of the company tries to answer the problems regarding the company's existence, the boundary between the company and the market, the organizational structure of the company, and the heterogeneity of the company's actions in the company's performance (Pamungkas and Utomo 2018). If there is a problem, management is likely to replace the auditor. Therefore, companies that change auditors are more likely to have a goal of fraudulent financial reporting. Based on this assumption and previous research by Nurcahyono et al. (2021), Utomo et al. (2019), it can be concluded that the frequent change of auditors implies fraud.

**Hypothesis 5 (H5).** *Auditor switching affects Fraudulent Financial Reporting.*

*2.6. Effect of Change in Director Affects Fraudulent Financial Reporting*

Fraud will not be carried out if the perpetrator cannot commit fraud. Therefore, position, intelligence, confidence, skill, effective deception, and stress management are all fundamental attributes of committing fraud. A change of director represents the capacity to articulate the ability to manage stress. Competence refers to a person's attitudes and skills that are very important in committing fraud, in addition to the possibility of someone acting fraudulently due to persuasion, coercion, or reasoning. Fraudsters must also know which door is open, representing an opportunity, and take advantage of it by passing it many times (Situngkir and Triyanto 2020).

Changes to the board of directors are only sometimes beneficial for the company. For example, a change of directors can be an attempt by the company to improve the performance of the previous directors by changing the composition of the board of directors or recruiting new directors who are considered more competent. On the other hand, the change in the company's directors may be an attempt to eliminate directors suspected of knowing about fraud in the company. Furthermore, new directors take time to adjust to the new culture, which limits performance effectiveness. This will result in a period of tension, which will increase the likelihood of fraud. Based on these assumptions and previous research by Achmad et al. (2022b), Evana et al. (2019), Sari et al. (2020), Utami and Pusparini (2019), it can be concluded that the change in directors has the potential to cause fraud.

**Hypothesis 6 (H6).** *Change in director affects Fraudulent Financial Reporting.*

*2.7. Effect of Change in Director Affects Fraudulent Financial Reporting*

A very arrogant attitude can lead to the possibility of fraud. Furthermore, because of his alleged superiority, the chief executive officer considers internal control irrelevant to him because of his status and position. CEOs will be seen as increasingly arrogant due to the increasing number of CEO images appearing in financial statements. The reason is the CEO's desire to show off the many strata he has in the organization to become more famous.

The CEO is expected to follow all company regulations and internal controls because of his role and position (Situngkir and Triyanto 2020).

The number of images of CEOs that often appear in the company's annual financial statements shows the frequency with which someone occupies the position of CEO. The frequency with which CEOs take photos is a representation of arrogance. Therefore, the number of CEO images displayed in the company's annual report can indicate the level of arrogance or superiority of the CEO when someone wants to show his rank, position, and presence in the organization (Ratnasari and Solikhah 2019). Based on these assumptions, it can be concluded that the more often the image of the CEO appears in the annual report, the higher the level of CEO arrogance, which will lead to fraud. Based on this approach, and in line with the research of Apriliana and Agustina (2017), Husmawati et al. (2017), Utami and Pusparini (2019), it is hypothesized that the frequent occurrence of CEO photos influences fraudulent financial reporting.

**Hypothesis 7 (H7).** *Arrogance affects Fraudulent Financial Reporting.*

*2.8. Effect of Collusion Affects Fraudulent Financial Reporting*

Collusion is a deceptive compact or agreement between two or more persons for one party to act on behalf of the other party for a negative purpose, such as defrauding a third party for personal gain (Vousinas 2019). Because there is a strong connection, the company will be able to get unique privileges and privileges that will increase the company's performance and value. Collusion has ties to the hexagon theory, where management can exploit the convenience and privilege of the company to conduct fraudulent financial reporting through manipulation. The misalignment of goals between agents and principals causes this manipulation. Agents seek to maximize the profits from their performance. Agents can commit fraud by using the resources provided by politicians. This is also related to adverse selection, which refers to the existence of information known by management but not shared with the principal (Lozano et al. 2016). Based on these assumptions, it can be concluded that the stronger the connections within a company, the higher the level of collusion that will lead to fraud. This assumption is in line with the research of Aviantara (2021), Cao et al. (2019), Wijayani and Ratmono (2020), leading to the hypothesis that collusion within a company influences fraudulent financial reporting.

**Hypothesis 8 (H8).** *Collusion affects Fraudulent Financial Reporting.*

**3. Methodology**

This study used a purposive sample technique in determining the sample used. The sample comprised banking companies listed on the Indonesia Stock Exchange (IDX) in 2017–2021 with the following criteria and explanations:

1. Banking companies listed on the Indonesia Stock Exchange (IDX) in 2017–2021.
2. Banking companies that published financial reports consecutively during 2017–2021.
3. Banking companies that were not listed in 2017–2021.

The analytical method used in this research is the quantitative data analysis method using the logistic regression analysis method with IBM SPSS 25(Chicago, Illinois, US) in data testing. The dependent variable ($Y$) is a dummy variable whose measurement uses the numbers 0 and 1. Hypothesis testing is performed using a *t*-test. The following regression model is used in this study to test the hypothesis:

$$Y_{i,t} = \alpha + \beta_1 x_{1i,t} + \beta_2 x_{2i,t} + \beta_3 x_{3i,t} + \beta_4 x_{4i,t} + \beta_5 x_{5i,t} + \beta_6 x_{6i,t} + \beta_7 x_{7i,t} + \beta_8 x_{8i,t} + \varepsilon_{i,t}$$

Information:

$Y$: Prediction of fraudulent financial reporting will occur, determined by a dummy variable where the number 1 denotes financial statements that are indicated to be fraudulent and 0 denotes a lack thereof.

1-$\beta$8: Regression coefficient for each independent variable
*X*1: Financial target
*X*2: Financial stability
*X*3: External pressure
*X*4: Ineffective monitoring
*X*5: Auditor switching
*X*6: Change in director
*X*7: Arrogance
*X*8: Collusion
*e*: Error

The research method used in this study is logistic regression analysis, which aims to determine the influence of a dependent variable on an independent variable. The raw data is first managed, classified, and tested using SPSS 25 software. The initial stage before testing on SPSS is to input all research variables into the SPSS program. Then, tests are carried out to produce output suitable for the analytical method.

The objects selected in this study are banking companies listed on the Indonesia Stock Exchange (IDX) in 2017–2021 that did not experience delisting during the study period. Based on the criteria for selecting a sample that has been determined from a total population of 49 banking companies listed on the Indonesia Stock Exchange (IDX) 2017–2021, a sample of 43 companies was obtained. Details of the research objects and samples are described in the following Table 1. And Research sample criteria in Table 2.

**Table 1.** Variable operational definition.

| Variable | Concept | Measurement | Scale | References |
|---|---|---|---|---|
| Fraudulent Financial Reporting | Fraudulent material misstatement of financial statements | F- score models | Ratio | (Saleh et al. 2021) |
| Financial Target | Financial targets/targets that must be achieved | Net profit after tax/Total assets | Ratio | (Manurung and Hardika 2015) |
| Financial Stability | Company's financial condition | Income/Total assets | Ratio | (Manurung and Hardika 2015) |
| External Pressure | Pressure from external parties on the company's internal parties | Total Debts/Total assets | Ratio | (Situngkir and Triyanto 2020) |
| Ineffective Monitoring | Ineffective company internal control | Number of independent commissioners/Number of commissioners | Ratio | (Husmawati et al. 2017) |
| Auditor Switching | Auditor change as a form of covering up fraudulent financial reporting committed (Utomo et al. 2019) | The dummy variable is coded one if there is a change of auditor and is given a code of 0 if there is no auditor replacement. | Nominal | (Utomo et al. 2019) |
| Change In Director | Change of the board of directors in a company | The dummy variable is coded one if there is a change of directors and is given a code of 0 if there is no change of directors. | Nominal | (Evana et al. 2019) |
| Arrogance | A person's selfish attitude in showing his power | Number of Chief Executive Officer photos shown in annual financial statements | Nominal | (Sawaka and Hiwa 2020) |
| Collusion | An agreement or cooperation between two or more parties that have the potential to commit fraud | The total number of commissioners who have concurrent positions | Nominal | (Vousinas 2019) |

Source: Data processed (2022).

**Table 2.** Research sample criteria.

| No | Criteria | Total |
|---|---|---|
| 1 | Banking companies listed on the Indonesia Stock Exchange in 2017–2021 | 49 |
| 2 | Banking companies that did not publish consecutive financial statements during 2017–2021 | 4 |
| 3 | Companies that experienced delisting during 2017–2021 | 2 |
| | Sample companies that meet the criteria | 43 |
| | Total research data (Total sample companies that meet the criteria during five years of research) | 215 |

Source: Data processed (2022).

## 4. Results

What follows are the results of data processed by SPSS 25, 2022, which includes tests, descriptive statistics, collinearity statistics, heteroscedasticity test results, Hosmer and Lemeshow test, and coefficient of determination (Nagelkerke's R-square). Logistics regression test results, the omnibus test of model coefficients (f-test), and WALD (Wald Chi-Squared Test) results.

The statistical test results in Table 3 show that the total sample consists of 215.

**Table 3.** Descriptive statistics.

| | N | Min | Max | Mean | Std. Dev | Variance | Skewness | Kurtosis |
|---|---|---|---|---|---|---|---|---|
| Fraud | 215 | −1.0 | 2.6 | 0.036 | 0.382 | 0.147 | 1.979 | 12.345 |
| ROA | 215 | −0.158 | 0.147 | 0.088 | 3.318 | 11.013 | −0.662 | 9.696 |
| SALTA | 215 | 0.041 | 0.290 | 0.094 | 0.039 | 0.002 | 2.653 | 8.722 |
| LEV | 215 | 0.137 | 1.257 | 0.785 | 0.167 | 0.028 | −2.143 | 6.893 |
| BDOUT | 215 | 0.250 | 0.750 | 0.558 | 0.102 | 0.010 | −0.381 | 0.257 |
| AUDCHANGE | 215 | 0 | 1 | 0.23 | 0.424 | 0.180 | 1.277 | −0.373 |
| DCHANGE | 215 | 0 | 1 | 0.39 | 0.489 | 0.239 | 0.457 | −1.812 |
| FREQCEOPIC | 215 | 1 | 25 | 4.55 | 3.543 | 12.553 | 2.710 | 9.511 |
| COLLUSION | 215 | 0 | 5 | 1.45 | 1.416 | 2.004 | 0.496 | −0.832 |
| Valid N (listwise) | 215 | | | | | | | |

Source: Data processed by SPSS 25, 2022; ROA: return on asset; LEV: leverage.

Based on Table 4, the regression analysis results show that the model's ability to predict fraudulent financial statements is 90.2%. From the table above, the possibility of a company committing financial statement fraud is 9.8% of the total sample of 215 data. Meanwhile, companies that did not commit fraud accounted for 90.2% of the total sample of 215 data. Furthermore, for the multicollinearity test results in Table 5, when the VIF value < 10 or the tolerance value > 0.01, then it is stated that there is no multicollinearity.

**Table 4.** Variable frequency statistics Y.

| | Frequency | Percent | Valid Percent | Cumulative Percent |
|---|---|---|---|---|
| 0 | 194 | 90.2 | 90.2 | 90.2 |
| Valid 1 | 21 | 9.8 | 9.8 | 9.8 |
| Total | 215 | 100.0 | 100.0 | 100.0 |

Source: Data processed by SPSS 25, 2022.

**Table 5.** Collinearity statistics.

| Model | | Tolerance | VIF |
|---|---|---|---|
| 1 | ROA | 0.698 | 1.433 |
| | SALTA | 0.633 | 1.581 |
| | LEVERAGE | 0.789 | 1.267 |
| | BDOUT | 0.925 | 1.082 |
| | AUDCHANGE | 0.953 | 1.050 |
| | DCHANGE | 0.985 | 1.015 |
| | FREQCEOPIC | 0.948 | 1.054 |
| | COLLUSION | 0.943 | 1.060 |

Source: Data processed by SPSS 25, 2022.

Coefficient Correlations are presented in Table 6, and collinearity diagnostics are presented in Table 7. and Table 8. The basis for decision-making in this test is that if the significance value is ≥0.05, it can be concluded that there is no heteroscedasticity problem, but vice versa. On the other hand, if the significance value is <0.05, it can be concluded that there is a heteroscedasticity problem. The above calculation results show that the variable significance value is more significant than 0.05. Therefore, the results of the heteroscedasticity test were obtained, as shown in Table 9. So, it can be concluded that there were no symptoms of heteroscedasticity in the regression model used.

**Table 6.** Coefficient Correlations.

| Model | | | COLLUSION | LEV | DCHANGE | AUDCHANGE | FREQCEOPIC | BDOUT | ROA | SALTA |
|---|---|---|---|---|---|---|---|---|---|---|
| 1 | Correlations | COLLUSION | 1.000 | −0.008 | −0.077 | 0.052 | −0.090 | 0.038 | −0.153 | 0.009 |
| | | LEVERAGE | −0.008 | 1.000 | 0.081 | −0.108 | −0.022 | −0.186 | −0.075 | 0.395 |
| | | DCHANGE | −0.077 | 0.081 | 1.000 | −0.027 | −0.029 | −0.008 | 0.026 | 0.040 |
| | | AUDCHANGE | 0.052 | −0.108 | −0.027 | 1.000 | 0.056 | −0.078 | 0.099 | −0.126 |
| | | FREQCEOPIC | −0.090 | −0.022 | −0.029 | 0.056 | 1.000 | 0.002 | −0.157 | 0.118 |
| | | BDOUT | 0.038 | −0.186 | −0.008 | −0.078 | 0.002 | 1.000 | 0.158 | −0.106 |
| | | ROA | −0.153 | −0.075 | 0.026 | 0.099 | −0.157 | 0.158 | 1.000 | −0.478 |
| | | SALTA | 0.009 | 0.395 | 0.040 | −0.126 | 0.118 | −0.106 | −0.478 | 1.000 |
| | Covariances | COLLUSION | 0.000 | −1.221 | −3.409 | 2.708 | −5.640 | 8.382 | −1.189 | 6.202 |
| | | LEVERAGE | −1.221 | 0.013 | 0.000 | −0.001 | −1.260 | −0.004 | −5.371 | 0.025 |
| | | DCHANGE | −3.409 | 0.000 | 0.001 | −3.919 | −5.090 | −5.268 | 5.662 | 0.001 |
| | | AUDCHANGE | 2.708 | −0.001 | −3.919 | 0.002 | 1.170 | −0.001 | 2.560 | −0.003 |
| | | FREQCEOPIC | −5.640 | −1.260 | −5.090 | 1.170 | 2.490 | 1.872 | −4.862 | 0.000 |
| | | BDOUT | 8.382 | −0.004 | −5.268 | −0.001 | 1.872 | 0.031 | 0.000 | −0.010 |
| | | ROA | −1.189 | −5.371 | 5.662 | 2.560 | −4.862 | 0.000 | 3.858 | −0.002 |
| | | SALTA | 6.202 | 0.025 | 0.001 | −0.003 | 0.000 | −0.010 | −0.002 | 0.294 |

Source: Data processed by SPSS 25, 2022.

**Table 7.** Collinearity diagnostics.

| Model | Dimension | Eigenvalue | Condition Index | (Constant) | ROA | SALTA | LEVERAGE |
|---|---|---|---|---|---|---|---|
| 1 | 1 | 5.846 | 1.000 | 0.00 | 0.00 | 0.00 | 0.00 |
| | 2 | 1.019 | 2.395 | 0.00 | 0.46 | 0.00 | 0.00 |
| | 3 | 0.719 | 2.851 | 0.00 | 0.14 | 0.00 | 0.00 |
| | 4 | 0.550 | 3.261 | 0.00 | 0.06 | 0.00 | 0.00 |
| | 5 | 0.405 | 3.800 | 0.00 | 0.02 | 0.01 | 0.00 |
| | 6 | 0.319 | 4.281 | 0.00 | 0.00 | 0.04 | 0.00 |
| | 7 | 0.106 | 7.426 | 0.00 | 0.27 | 0.55 | 0.09 |
| | 8 | 0.026 | 15.071 | 0.00 | 0.04 | 0.15 | 0.47 |
| | 9 | 0.010 | 23.693 | 0.99 | 0.01 | 0.26 | 0.44 |

Source: Data processed by SPSS 25, 2022.

**Table 8.** Collinearity diagnostics.

| Model | Dimension | BDOUT | AUDCHANGE | DCHANGE | FREQCEOPIC | COLLUSION |
|-------|-----------|-------|-----------|---------|------------|-----------|
| 1 | 1 | 0.00 | 0.01 | 0.01 | 0.01 | 0.01 |
| | 2 | 0.00 | 0.14 | 0.01 | 0.00 | 0.01 |
| | 3 | 0.00 | 0.68 | 0.09 | 0.01 | 0.02 |
| | 4 | 0.00 | 0.03 | 0.87 | 0.03 | 0.01 |
| | 5 | 0.00 | 0.05 | 0.00 | 0.05 | 0.93 |
| | 6 | 0.00 | 0.08 | 0.01 | 0.80 | 0.00 |
| | 7 | 0.01 | 0.00 | 0.00 | 0.07 | 0.00 |
| | 8 | 0.70 | 0.00 | 0.00 | 0.00 | 0.00 |
| | 9 | 0.28 | 0.01 | 0.01 | 0.02 | 0.01 |

Source: Data processed by SPSS 25, 2022.

**Table 9.** Heteroscedasticity test results.

| Model | Unstandardized Coefficients B | Standardized Coefficients Std. Error | Beta | t | Sig. |
|-------|-------------------------------|--------------------------------------|------|-----|------|
| (Constant) | −3.587 | 1.386 | | −2.588 | 0.011 |
| ROA | −0.160 | 0.059 | −0.188 | −2.702 | 0.058 |
| SALTA | 19.975 | 5.179 | 0.282 | 3.857 | 0.050 |
| LEVERAGE | −6.886 | 1.107 | −0.407 | −6.222 | 0.050 |
| BDOUT | 0.974 | 1.675 | 0.035 | 0.581 | 0.562 |
| AUDCHANGE | 1.680 | 0.398 | 0.251 | 4.220 | 0.071 |
| DCHANGE | 1.416 | 0.339 | 0.244 | 4.175 | 0.083 |
| FREQCEOPIC | 0.047 | 0.048 | 0.059 | 0.992 | 0.323 |
| COLLUSION | −0.298 | 0.120 | −0.149 | −2.485 | 0.064 |

Source: Data processed by SPSS 25, 2022.

From Table 10, the regression analysis results showed that the Hosmer and Lemeshow goodness of fit test obtained a significance of 0.894. Therefore, the test results show that if the probability value (*p*-value) $\geq$ 0.05 (significant value), namely 0.894 $\geq$ 0.05, then hypothesis 0 is accepted. This indicates that there is no significant difference between the model and the data, so the regression model in this study is feasible and able to predict the observed value.

**Table 10.** Hosmer and Lemeshow test.

| Step | Chi-Square | Df | Sig. |
|------|------------|-----|------|
| 1 | 3.564 | 8 | 0.894 |

Source: Data processed by SPSS 25, 2022.

The results of the regression analysis are in Table 11. The coefficient of determination, as seen from the Nagelkerke R-square value, is 0.498. This indicates that the ability of the independent variable to explain the dependent variable is 49.8%. In contrast, the rest is explained by other variables outside of this research model, which is equal to 50.2%.

**Table 11.** Coefficient of determination (Nagelkerke's R-square).

| Step | −2 Log Likelihood | Cox and Snell R-Square | Nagelkerke R-Square |
|------|-------------------|------------------------|---------------------|
| 1 | 45.825 | 0.189 | 0.498 |

Source: Data processed by SPSS 25, 2022.

Table 12 above shows the results of the logistic regression test with the following model:

$$Y = 4.689 - 0.095X1 + 8.669X2 - 6.231X3 - 1.201X4 + 0.799X5 + 0.887X6 - 1.380X7 - 0.001X8$$

**Table 12.** Logistics regression test results.

|  | B | S.E | Wald | Df | Sig. | Exp (B) |
|---|---|---|---|---|---|---|
| Step 1 an X1 | −0.095 | 0.104 | 0.833 | 1 | 0.361 | 0.909 |
| X2 | 8.669 | 10.508 | 0.681 | 1 | 0.409 | 5817.570 |
| X3 | −6.231 | 1.990 | 9.803 | 1 | 0.002 | 0.002 |
| X4 | −1.201 | 5.052 | 0.057 | 1 | 0.812 | 0.301 |
| X5 | 0.799 | 0.894 | 0.800 | 1 | 0.371 | 2.224 |
| X6 | 0.887 | 0.846 | 1.097 | 1 | 0.295 | 2.427 |
| X7 | −1.380 | 0.546 | 6.391 | 1 | 0.011 | 0.251 |
| X8 | 0.001 | 0.350 | 0.000 | 1 | 0.999 | 1.001 |
| Constant | 4.689 | 3.345 | 1.965 | 1 | 0.161 | 108.703 |

a. Variable(s) entered on step 1: X1, X2, X3, X4, X5, X6, X7, X8. Source: SPSS 25 output, 2022.

### 4.1. Omnibus Test of Models Coefficients (f-Test)

Based on Table 13 with a sample of ($n$ = 215) and the number of independent and dependent variables (k = 9), then the degree of freedom (df1) = k − 1 = 9 − 1 = 8 and (df2) = $n$ − k = 215 − 9 = 206, where the level of significance = 0.05. Then the F table can be calculated using the Ms Excel formula with the insert function formula as follows:

$$\text{F table} = \text{FINV (Probability,deg\_freedom1,deg\_freedom2)}$$

$$\text{F table} = \text{FINV (0.05,8,163)}$$

$$\text{F table} = 1.995605$$

**Table 13.** Simultaneous test results.

|  | Chi-Square | df | Sig. |
|---|---|---|---|
| Step 1 Step | 35.947 | 8 | 0.000 |
| Block | 35.947 | 8 | 0.000 |
| Model | 35.947 | 8 | 0.000 |

Source: SPSS 25 output, 2022.

Based on the results, the F count value is greater than the F table (35.947 > 1.995605) with a significance level (0.000 < 0.05), and it can be concluded that the independent variables simultaneously affect the dependent variable.

### 4.2. Impact of Financial Targets on Fraudulent Financial Reporting

In this study, based on Table 14, the results of hypothesis testing 1 (H1), in which the financial target is proxied by the return on assets (ROA), are rejected. It can be interpreted that financial targets do not affect fraudulent financial reporting. The value of count (wald) is smaller than the table (0.833 < 1.974625), and the probability value is higher than the significance value (0.361 > 0.05). This is what underlies the statement that H1 is rejected. Based on this, a high or low ROA value cannot be used as a benchmark for fraudulent financial reporting. Management does not react negatively to high or low targets. An increase in ROA does not always indicate that the company is committing fraud. However, it could be due to an increase in the quality of operations and the recruitment of qualified employees. The company believes in investing in the modernization of enterprise information systems, efficiency in business processes at a higher cost than benefit, and implementing policies to meet the stated goals. As a result, when the company's targets are raised, management will not feel pressured. This is because the ROA can be used to measure performance. If the targeted ROA is still reasonable and achievable, then the ROA

will not be a trigger for fraud. ROA, which is a proxy for the financial target variable, which in the fraud hexagon theory is an element of the stimulus factor, is stated not to support the theory.

**Table 14.** WALD test results.

|  | B | S.E. | Wald | df | Sig. | Exp(B) |
|---|---|---|---|---|---|---|
| Step 1 X1 | −0.095 | 0.104 | 0.833 | 1 | 0.361 | 0.909 |
| X2 | 8.669 | 10.508 | 0.681 | 1 | 0.409 | 5817.570 |
| X3 | −6.231 | 1.990 | 9.803 | 1 | 0.002 | 0.002 |
| X4 | −1.201 | 5.052 | 0.057 | 1 | 0.812 | 0.301 |
| X5 | 0.799 | 0.894 | 0.800 | 1 | 0.371 | 2.224 |
| X6 | 0.887 | 0.846 | 1.097 | 1 | 0.295 | 2.427 |
| X7 | −1.380 | 0.546 | 6.391 | 1 | 0.011 | 0.251 |
| X8 | 0.001 | 0.350 | 0.000 | 1 | 0.999 | 1.001 |
| Constant | 4.689 | 3.345 | 1.965 | 1 | 0.161 | 108.703 |

Source: SPSS 25 output, 2022.

From the research results, it is known that Bank Rakyat Indonesia Argoniaga Tbk. Obtained the highest ROA value of 14.75, followed by an F score value that is not classified as an indication of fraudulent financial reporting, the value obtained being 0.6751. On the other hand, Bank Jago Tbk. has the lowest ROA value of −15.89, followed by the F score of 0.9687, which is classified as an indication of fraudulent financial reporting. Therefore, this study cannot support the fraud hexagon theory, which states that financial stability is a stimulus element influencing fraud. Instead, they state that financial targets do not affect fraudulent financial reporting. However, this contradicts the research of (Akbar 2017; Sari Pramono et al. 2020), according to which financial targets influence fraudulent financial reporting.

*4.3. Impact of Financial Stability on Fraudulent Financial Reporting*

In Table 15, for the variable X2 (financial stability), which is proxied by SALTA, the results of the count are smaller than the t-table (0.681 < 1.974625). The probability value obtained is greater than the level of significance (0.409 > 0.05). These results indicate that H2, which states that financial stability affects fraudulent financial reporting, must be rejected. This study shows that financial stability does not affect the occurrence of fraudulent financial reporting. This is because when financial conditions are unstable or disrupted, managers in the sample companies do not necessarily manipulate financial statements to increase the company's attractiveness to external parties when the company's average growth is below average. After all, this will worsen the situation. Another possibility is that the sample companies are well supervised by the board of commissioners so that when managers are under pressure due to threatened financial conditions, the occurrence of fraudulent financial reporting is not affected.

From the study results, it is known that Bank BTPN Syariah Tbk obtained the highest SALTA value of 0.29, followed by an F score value which is not classified as an indication of fraudulent financial reporting, the value obtained being 0.2670. Therefore, this study cannot support the fraud hexagon theory, which states that financial stability is a stimulus element influencing fraud. This study's results align with previous research conducted by (Manurung and Hardika 2015). They state that financial stability does not affect fraudulent financial reporting. However, it contradicts the research of Utami and Pusparini (2019), which states that financial stability influences fraudulent financial reporting.

**Table 15.** Summary of hypotheses.

| Hypotheses | Wald | Sign | Conclusion |
|---|---|---|---|
| H1: Financial target affects fraudulent financial reporting | 0.8330 | 0.361 | Rejected |
| H2: Financial stability affects fraudulent financial reporting | 0.681 | 0.409 | Rejected |
| H3: External pressure affects fraudulent financial reporting | 9.803 | 0.002 | Accepted |
| H4: Ineffective monitoring affects fraudulent financial reporting | 0.057 | 0.812 | Rejected |
| H5: Auditor switching affects fraudulent financial reporting | 0.800 | 0.371 | Rejected |
| H6: Change in director affects fraudulent financial reporting | 1.097 | 0.295 | Rejected |
| H7: Arrogance affects fraudulent financial reporting | 6.391 | 0.011 | Accepted |
| H8: Collusion affects fraudulent financial reporting | 0.000 | 0.999 | Rejected |

Source: Data processed (2022).

### 4.4. Impact of External Pressure on Fraudulent Financial Reporting

In this study, hypothesis 3 (H3) test results, in which external pressure is proxied by leverage, are accepted. It can be interpreted that external pressure affects fraudulent financial reporting. The value of the t-count (wald) is greater than the t-table (9.803 > 1.974625), and the probability value is lower than the significance value (0.002 < 0.05). This underlies H3, which states that external pressure affects acceptable fraudulent financial reporting. External pressures from high credit risk due to large debts encourage management to manipulate financial statements to persuade creditors. The higher the level of leverage, the greater the likelihood of fraudulent financial reporting. Management pressure to obtain additional funds encourages management to do everything possible, including falsifying financial statements. Next, the lender will consider various factors affecting whether to apply for a loan. Naturally, creditors will approve loans from companies with a credible and positive reputation. Consequently, the greater the external pressure, the greater the potential for management to engage in fraudulent financial reporting.

The external pressure variable is proxied by leverage, an element of the stimulus in the fraud hexagon theory. The research results show that Industrial Bank of Korea (IBK) Indonesia (Tbk) obtained the highest leverage value of 1.257, followed by an F score of 1.1020, which is classified as an indication of fraudulent financial reporting. On the other hand, Bank Syariah Indonesia Tbk. has the lowest leverage value of 0.1371, followed by an F score of 1.4957, which is classified as an indication of fraudulent financial reporting. This proves that the higher the company's leverage ratio, the higher the risk of management committing fraudulent financial reporting. Therefore, this study supports the fraud hexagon theory, which states that external pressure is a stimulus element influencing fraud. The results of this study are in line with previous research conducted by Achmad et al. (2022a), Pamungkas and Utomo (2018), Situngkir and Triyanto (2020), Utami and Pusparini (2019). They state that external pressure influences fraudulent financial reporting. However, this contradicts the research of Achmad et al. (2022b), which states that external pressure does not affect fraudulent financial reporting.

### 4.5. Impact of Ineffective Monitoring on Fraudulent Financial Reporting

In this study, the results of hypothesis testing 4 (H4), in which ineffective monitoring was proxied by BDOUT, were rejected. It can be interpreted that ineffective monitoring does not affect fraudulent financial reporting. The value of the t-count (wald) is smaller than the t-table (0.57 < 1.974625), and the probability value is higher than the significance value (0.812 > 0.05). This is what underlies the statement that H4 is rejected. Based on this, a high or low BDOUT value cannot be used as a benchmark for fraudulent acts in financial statements. Many or only some independent commissioners are powerless to prevent fraudulent financial reporting. The number of independent commissioners may be only a regulatory requirement for good corporate governance; in practice, they can still be influenced by the corporate intervention. The presence of an independent board

of commissioners provides a little guarantee of oversight of the company. The board of commissioners is responsible for ensuring the implementation of the company's strategy, oversees management, and requires accountability. However, the increase in independent commissioners has not impacted the company's operational supervision because, if the board of commissioners intervened, the company's supervision would be one-sided.

Based on the research, it is known that Bank Neo Commerce Tbk. Obtained the highest BDOUT value of 0.75, followed by an F score of 0.1550, which is not classified as an indication of fraudulent financial reporting. Therefore, this study cannot support the fraud hexagon theory, which states that ineffective monitoring is an element of opportunity to influence fraud. This study's results align with previous research conducted by (Nanda et al. 2019). They stated that ineffective monitoring did not affect fraudulent financial reporting.

### 4.6. Impact of Auditor Switching on Fraudulent Financial Reporting

In Table 3, for the X5 variable (auditor switching), which is proxied by the dummy variable, the results of the t-count are smaller than the t-table (0.800 < 1.974625). The probability value obtained is greater than the significance level (0.371 > 0.05). These results underlie the rejection of H5, which states that auditor switching affects fraudulent financial reporting. Based on the findings of auditor switching analysis, it does not affect the potential for fraudulent financial reporting, which indicates that the company's auditor turnover does not cause the company to commit fraudulent actions. The effect of auditor turnover on the potential for fraudulent financial reporting is due to the possibility of a company changing auditors. Based on the completion of a predetermined contract or other matters. Auditor turnover is caused by reducing the company's audit fees by improving corporate governance so that the following audit fee is manageable. Of course, this does not authorize the auditors of a company to commit fraud.

The analysis results in this study were 40 companies changing auditors during the research period. However, of the 40 companies, only four companies PT Bank IBK Indonesia Tbk (AGRS), PT Bank Amar Indonesia Tbk. (AMAR), Bank Capital Indonesia Tbk. (BACA), Indications of fraudulent financial reporting marked PT. Bank Jtrust Indonesia Tbk (BCIC). The conclusion drawn from these results is that auditor switching, an element of rationalization, does not support the statement in the fraud hexagon theory. This study's results align with previous research conducted by (Nurcahyono et al. 2021; Utomo et al. 2019). They state that auditor switching does not affect fraudulent financial reporting.

### 4.7. Impact of a Change in Director on Fraudulent Financial Reporting

In Table 3 for the variable X6 (change in director), which is proxied by the dummy variable, the results of the count are smaller than the t-table (1.097 < 1.974625). The probability value obtained is greater than the level of significance (0.295 > 0.05). These results underlie the statement that H6, which states that change in director affects fraudulent financial reporting, is rejected. The findings of this study indicate that management does not use the change of directors to commit fraud. The less frequent the change of directors. The better the ability of the directors to manage the company and maintain it. A change of directors in a company can occur for various reasons, including the resignation of the directors, the death of the old directors, and the need for new directors to fill vacancies. A desire to improve the company's performance and quality by recruiting directors considered better or more capable than before.

The analysis results in this study were that 40 companies changed auditors during the research period. However, of the 40 companies, only 4 companies PT Bank IBK Indonesia Tbk (AGRS), PT Bank Amar Indonesia Tbk. (AMAR), Bank Capital Indonesia Tbk. (BACA), PT. Bank Jtrust Indonesia Tbk (BCIC). So, a change in director, an element of capability, cannot support the fraud hexagon theory. This study's results align with previous research by (Situngkir and Triyanto 2020). They stated that changes in directors did not affect fraudulent financial reporting.

*4.8. Impact of Arrogance on Fraudulent Financial Reporting*

In this study, hypothesis testing 7 (H7) results, in which arrogance is proxied by the frequency of many CEO photos in the annual financial statements, are accepted. It can be interpreted that arrogance affects fraudulent financial reporting. The value of the t-count (wald) is greater than the t-table (6.391 > 1.974625), and the probability value is lower than the significant value (0.011 < 0.05). This underlies H7, which states that arrogance affects acceptable fraudulent financial reporting. The more photos of the CEO presented in the company's annual report that show the arrogance of the CEO in a company, the higher the possibility of fraud due to the CEO's arrogance and superiority. So he feels that internal control does not apply to him personally because of his status and position. The research results show Bank Commerce International Merchant Bankers (CIMB) Niaga Tbk. Obtained the highest frequency of 25, followed by an F score of 0.8377, which is classified as an indication of fraudulent financial reporting. This research can support the fraud hexagon theory, which states that the frequency of the CEO's photo in the financial statements is an element of arrogance that affects fraud. This study's results align with previous research by (Apriliana and Agustina 2017). They state that arrogance influences fraudulent financial reporting.

*4.9. Collusion on Fraudulent Financial Reporting*

In Table 4, for the variable X8 (collusion), which is proxy by the number of independent commissioners who have multiple positions, the results of the t-count are smaller than the t-table (0.000 < 1.974625). The probability value obtained is greater than the level of significance (0.999 > 0.05). These results underlie the statement that H8, which states that collusion affects fraudulent financial reporting, is rejected. Concurrent positions on the independent board of commissioners do not make them non-independent in corporate governance, but each independent commissioner is relatively independent. This is supported by concurrent positions of independent commissioners in the object of research that do not violate Law Number 19 of 2003 concerning state-owned enterprises or the financial services authority (OJK). Moreover, concurrent positions are only dominated by one of the criteria, namely as a former official or former military. The current research results show that the highest total concurrent positions range from 3 to 5. However, when considering all companies with several concurrent positions on the board of commissioners, whether 3, 4, or 5, the F score needs to be followed by an indication of fraudulent financial reporting. This confirms that collusion cannot support the fraud hexagon theory. This study's results align with previous studies by (Vousinas 2019).

**5. Conclusions**

This study aims to prove empirically the influence of stimulus (financial target, financial stability, and external pressure), capability (change in director), collusion (total board of commissioners who have multiple positions), opportunity (ineffective monitoring), rationalization (auditor switching), and arrogance (frequency of the number of CEO photos in the annual financial statements) on fraudulent financial reporting in banking companies listed on the Indonesia Stock Exchange (IDX) in 2017–2021. Based on the analysis that has been carried out, the conclusions that can be drawn are as follows: external pressure and arrogance affect fraudulent financial reporting. However, financial targets, financial stability, ineffective monitoring, auditor switching, external pressure, change in director, and collusion do not affect fraudulent financial reporting. For a company to have a system for preventing the occurrence of embezzlement, it must put in place a system of detection, monitoring, and systems review policies in the field of human resources (HR). Therefore, the work unit function that manages human resources is a significant factor in creating and implementing optimal anti-fraud policies. Some policies and procedures for human resources that must be in place include the process of employee recruitment, education, transparency rotation process, mutation-promotion, sanctioning, eliminating discriminatory policies (not assertive), giving rewards, integrity, remuneration, and performance system.

## 6. Suggestions

Future researchers are expected to be able to use other proxies that are more varied in using the hexagon fraud model. For opportunity variables, institutional ownership, the quality of external and audit commissioners in the audit committee can be used, and for capability, the quality of CEOs in the broader population; further research can also add moderating variables to improve the accuracy of research results.

**Author Contributions:** Conceptualization, T.A., M.R.A.H. and I.D.P.; methodology, I.G., D.I.H.; validation, T.A.; formal analysis, I.G., T.A. and D.I.H.; investigation, M.R.A.H., and I.D.P.; resources, I.D.P.; data curation, D.I.H. and I.D.P.; writing—original draft preparation, TA. and I.D.P.; project administration, I.D.P.; funding acquisition, D.I.H. and M.R.A.H.; writing—review and editing, T.A. and I.D.P. All authors have read and agreed to the published version of the manuscript.

**Funding:** This study was funded by Research Funded Research of International Publications of High Reputation (RPIBT) Universitas Diponegoro, grant number No. SPK: 233-34/UN7.6.1/PP/2022, and the APC was funded by the Directorate of Research and Community Service, Ministry of Education, Culture, Research, and Technology, Indonesia.

**Informed Consent Statement:** Not applicable.

**Data Availability Statement:** Not applicable.

**Acknowledgments:** We would like to thank the Research of International Publications of High Reputation (RRPIBT) No. SPK: 233-34/UN7.6.1/PP/2022.

**Conflicts of Interest:** The authors declare no conflict of interest.

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
