# Peer review of "Detecting Fraudulent Financial Reporting Using the Fraud Hexagon Model: Evidence from the Banking Sector in Indonesia"

_economies, doi:10.3390/economies11010005_

Round 1

Reviewer 1 Report

Dear authors, congratulations on the article. I find it interesting and very well structured. I love how you found the hypotheses. Connection with literature and high-level argumentation.

I believe that the research methodology and results are also very well presented.

But I miss a part

1. discussion - after the results, I would like you to discuss your results in the context of other works

2. in conclusions - please write the directions for further research

3. At the end of the introduction, it is worth writing what the job structure is.

Congratulations on your great work

Author Response

Reviewer 1

Thank you for the recommendations and reviews for our articles.

In the discussion section - after the results, we have complemented it with supporting research support.

In conclusion, we have added instructions for further research as follows. In order for the company to have a system to prevent the occurrence of embezzlement, the company makes a system of detection, monitoring, and systems that review policies in the field of human resources (HR). Therefore, the work unit function who manages human resources or human resources has a very important factor in creating optimization of the implementation of anti-fraud policies. Some policies and procedures Human resources that must be held at least include the process of employee recruitment, education, transparency rotation process, mutation-promotion, sanctioning, eliminating biased policies (not assertive), giving rewards, integrity, remuneration, and performance system.

At the end of the introduction, we write down what is the structure of the work: This study is divided into five main sections. The first is an introduction which contains the phenomenon of GAP in the banking sector in Indonesia and Research GAP on Fraudulent Financial Reporting. In Section 2, a literature review is presented with an explanation of the eight factors of Fraud Hexagon on Fraudulent Financial Reporting. This section also includes Hypothesis development. In Section 3, the Methodology covers Variable Operational Definitions and Research Sample Criteria. Section 4 provides the results and in Section 5, the discussion and conclusions of the study are explained.

Thank you so much for the time and review that has been given to improve our article. We are very grateful for the guidance, knowledge and extraordinary things to improve our articles so that they become even better.

Best regard

Reviewer 2 Report

Review of the Manuscript economies - 2005576  „Detecting fraudulent financial reporting using fraud hexagon model: evidence from banking sector in Indonesia” for the Economies.

General Comments

From my point of view, it is a very interesting topic and simultaneously it seems that to the best of my knowledge is the first empirical research which study the role of the Detecting fraudulent financial reporting using fraud hexagon model: evidence from banking sector in Indonesia

The paper consists of the following sections: Introduction, Literature Review, Methodology, Results, Conclusions and Suggestions.

However, I find some recommendations:

1.       It would be very useful to add in the "Introduction" section the purpose, objectives and hypothesis of the research.

2.       I recommend the authors to make a complete descriptive analysis and to include a series of indicators and tests such as standard deviation, Jarque-Berra, Kurtosis, probabilities, etc., and the number of observations taken in the sample.

3.       It is very important that the authors present the correlation matrix and the covariance matrix and explain the results obtained.

4.       I think that the authors should present the VIF test to verify heteroskedasticity and also to verify endogeneity.

5.       At the same time, the authors must explain why they chose cross-section with fixed effects and not with random effects. That is why the authors must do the Hausman test.

6.       Also,  we consider the literature is not enough and that is why, we recommend the authors to refer to other recent works indexed in Web of Science, Scopus, Emerald, Cambridge, and of course MDPI Journals. We suggest that the authors cite papers published in MDPI journals and Web of Science Journals, such as:

  1. Batrancea L.M., Pop MC, Rathnaswamy, M.M., Batrancea I., Rus M-I,  (2021), An Empirical Investigationon the Transition Process toward a Green Economy, Sustainability, 13(23):13151. https://doi.org/10.3390/su132313151.

2.       Batrancea, L., Rathnaswamy, M.M., MI Rus, Tulai H., (2022), Determinants of Economic Growth for the Last Half Century: A Panel Data Analysis on 50 Countries, Journal of the Knowledge Economy. https://doi.org/10.1007/s13132-022-009944-9.

The conclusions must be expanded with possible economic policy implications of the research undertaken.

All in all, I consider that the paper must be improved.

Author Response

Reviewer 2

General comments

In the "Introduction" section we add the aims, objectives and research hypotheses as follows: The novelty of this research is to test the Fraud Hexagon which is the latest fraud model with eight factors by adding a collusion variable with a proxy using the total number of commissioners who have concurrent positions to analyze so as to prevent Fraudulent Financial Reporting in the banking sector in Indonesia. The purpose of this study is to examine the potential of Fraudulent Financial Reporting using the Fraud Hexagon Theory such as stimulus (Financial Target, Financial Stability, and External Pressure), Capability (Change of Directors), Collusion (Number of Board of Commissioners who have multiple positions), Opportunity (Unmonitored Effective), Rationalization (Auditor Switching), and Arrogance (Frequency of the number of CEO photos in the annual financial report) have an effect on Fraudulent Financial Reporting in the banking sector in Indonesia.

  1. This study is divided into five main sections. The first is an introduction which contains the phenomenon of GAP in the banking sector in Indonesia and Research GAP on Fraudulent Financial Reporting. In Section 2, a literature review is presented with an explanation of the eight factors of Fraud Hexagon on Fraudulent Financial Reporting. This section also includes Hypothesis development. In Section 3, the Methodology covers Variable Operational Definitions and Research Sample Criteria. Section 4 provides the results and in Section 5, the discussion and conclusions of the study are explained.
  2. I recommend the author to make a complete descriptive analysis and include a series of indicators and tests such as standard deviation, Jarque-Berra, Kurtosis, probability, etc., and the number of observations taken in the sample. We have added them to Table 3. Descriptive Statistics.
  3. It is very important for the author to present the correlation matrix and covariance matrix and explain the results obtained.

We have added them to Table 5. Collinearity Statistics and Table 6. Coefficient Correlations

  1. For the heteroscedasticity test we have added it to Table 9. Heteroscedasticity Test Results The results of the heteroscedasticity test were obtained as shown in Table 9. So it can be concluded that there were no symptoms of heteroscedasticity in the regression model used.
  2. From Table 10. the results of the regression analysis showed that the results of the Hosmer and Lemeshow Goodness of Fit Test obtained a significance of 0.894. The test results show that the probability value (P-value) ≥ 0.05 (significant value), namely 0.894 ≥ 0.05, then H0 is accepted. This indicates that there is no significant difference between the model and the data so that the regression model in this study is feasible and able to predict the observed value.
  3. Thank you for the references that have been provided. We have added references to other recent works indexed in the Web of Science, Scopus, Emerald, Cambridge, and of course the MDPI Journal. Such as Batrancea L.M., Pop MC, Rathnaswamy, M.M., Batrancea I., Rus M-I, (2021), An Empirical Investigation of the Transition Process Towards a Green Economy, Sustainability, 13(23)::13151. https://doi.org/10.3390/su132313151 and Batrancea, L., Rathnaswamy, M.M., MI Rus, Tulai H., (2022), Determinants of Economic Growth in the Last Half Century: Analysis of Panel Data in 50 Countries, Journal of Knowledge Economy. https://doi.org/10.1007/s13132-022-009944-9.

We have added and expanded the conclusions with possible economic policy implications from the research conducted, namely: Future researchers are expected to be able to use other proxies that are more varied in using the Hexagon fraud model. For opportunity variables, institutional ownership, Quality of external and audit Commissioners in the audit commit- tee can be used and for Capability can use Quality of CEO's in a wider population and further research can also add moderating variables to improve research results to be more accurate.

Thank you so much for the time and review that has been given to improve our article. We are very grateful for the guidance, knowledge and extraordinary things to improve our articles so that they become even better.

Best regard

Reviewer 3 Report

My comments:

*The abstract is not informative. 

*The introduction section does not clearly explain the scientific contribution of the study. 

*Literature review sections should explain the gap in literature.

*The authors should explain the reasons for the selected methods. 

*The definition of Fraudulent Financial Reporting should be clearer. 

*The results should be backed by the literature. 

*The conclusion should be revised. 

*I cannot see the comprehensive policy recommendations. 

Author Response

Reviewer 3

My comments:

* Abstract is not informative.

Abstract: The purpose of this study is to examine the potential for Fraudulent Financial Reporting using the Fraud Hexagon Theory such as stimulus (Financial Target, Financial Stability, and External Pressure), Capability (Change in Director), Collusion (Total Board of Commissioners who have multiple positions), Opportunity (Ineffective Monitoring), Rationalization (Auditor Switching), and Arrogance (Frequency of the number of photos of the CEO in the annual financial statements) affect Fraudulent Financial Reporting. The sample of this study is Banking companies listed on the Indonesia Stock Exchange (IDX) in 2017-2021 with a total sample of 215 and data processing using SPSS 25 software. The results of this study indicate that External Pressure, Arrogance affects Fraudulent Financial Reporting. However, Financial targets, Financial Stability, Ineffective Monitoring, Auditor Switching, Change In Director and Collusion do not affect Fraudulent Financial Reporting. In order for the company to have a system to prevent the occurrence of embezzlement, the company created a system of detection, monitoring, and systems that review policies in the field of human resources (HR).

*The introduction does not clearly explain the scientific contribution of this research.

The novelty of this research is to test the Fraud Hexagon which is the latest fraud model with eight factors by adding a collusion variable with a proxy using the total number of commissioners who have concurrent positions to analyze so as to prevent Fraudulent Financial Reporting in the banking sector in Indonesia. The purpose of this study is to examine the potential of Fraudulent Financial Reporting using the Fraud Hexagon Theory such as stimulus (Financial Target, Financial Stability, and External Pressure), Capability (Change of Directors), Collusion (Number of Board of Commissioners who have multiple positions), Opportunity (Unmonitored Effective), Rationalization (Auditor Switching), and Arrogance (Frequency of the number of CEO photos in the annual financial report) have an effect on Fraudulent Financial Reporting in the banking sector in Indonesia.

*The author must explain the reasons for the method chosen.

The analytical method used in this research is the quantitative data analysis method using the logistic regression analysis method with IBM SPSS 25 in data testing because the dependent variable (Y) is a dummy variable whose measurement uses the numbers 0 and 1. Hypothesis testing is done by t-test.

*Definition of Fraudulent Financial Reporting should be clearer.

Fraudulent Financial Reporting is Fraudulent material misstatement of financial statements (Alsinglawi, Mahmoud, and Saleh 2021)

*Results should be supported by literature.

The results of this study are in line with previous research conducted by Achmad, Hapsari, and Pamungkas (2022); Pamungkas and Utomo (2018); Situngkir and Triyanto (2020); Utami and Pusparini (2019). They state that external pressure influences Fraudulent Financial Reporting. However, this contradicts the research of Larum et al., (2021) and Achmad et al., (2022) which state that external pressure does not affect Fraudulent Financial Reporting.

The conclusion drawn from these results is that auditor switching which is an element of rationalization does not support the statement in the fraud hexagon theory. The results of this study are in line with previous research conducted by (Nurcahyono et al. 2021; Utomo et al. 2019). They state that auditor switching does not affect Fraudulent Financial Reporting.

The results of this study are in line with previous research conducted by (Apriliana and Agustina 2017). They state that arrogance influences Fraudulent Financial Reporting.

*Conclusions should be revised.

In order for the company to have a system to prevent the occurrence of embezzlement, the company created a system of detection, monitoring, and systems that review policies in the field of human resources (HR). Therefore, the work unit function who manages human resources or human resources has a very important factor in creating optimization of the implementation of anti-fraud policies. Some policies and procedures Human resources that must be held at least include the process of employee recruitment, education, transparency rotation process, mutation-promotion, sanctioning, eliminating biased policies (not assertive), giving rewards, integrity, remuneration, and performance system.

*I can't see comprehensive policy recommendations.

Suggestions

Future researchers are expected to be able to use other proxies that are more varied in using the Hexagon fraud model. For opportunity variables, institutional ownership, Quality of external and audit Commissioners in the audit commit- tee can be used and for Capability can use Quality of CEO's in a wider population and further research can also add moderating variables to improve research results to be more accurate .

Thank you so much for the time and review that has been given to improve our article. We are very grateful for the guidance, knowledge and extraordinary things to improve our articles so that they become even better.

Best regard

Reviewer 4 Report

The authors propose an interesting approach and the analysis is soundly conducted.

The abstract is well written and summarizes the key points of the paper. However, in the Introduction the authors miss to outline the novelty/originality of their research approach, the literature gap they intend to fill, the added value of their research: did they employ a new method for this strand of literature, did they use new variables ?. They should provide more details about these issues, in order to attract the interest of potential readers. 

Section 4 starts abruptly, with a series of tables. To increase the flow of ideas and facilitate readers' understanding it is recommended to add 1-2 paragraphs before the first table in this section, then to explain the results of the descriptive stats table, and so on. Try to introduce each table, by adding a phrase mentioninf it.

The concluding section should be developed more, it is very short and expeditive.

Author Response

Reviewer 4

The author proposes an interesting approach and the analysis is well done.

The abstract has been completed and is as follows:

The purpose of this study is to examine the potential for Fraudulent Financial Reporting using the Fraud Hexagon Theory such as stimulus (Financial Target, Financial Stability, and External Pressure), Capability (Change in Director), Collusion (Total Board of Commissioners who have multiple positions), Opportunity (Ineffective Monitoring), Rationalization (Auditor Switching), and Arrogance (Frequency of the number of photos of the CEO in the annual financial statements) affect Fraudulent Financial Reporting. The sample of this study is Banking companies listed on the Indonesia Stock Exchange (IDX) in 2017-2021 with a total sample of 215 and data processing using SPSS 25 software. The results of this study indicate that External Pressure, Arrogance affects Fraudulent Financial Reporting. However, Financial targets, Financial Stability, Ineffective Monitoring, Auditor Switching, Change In Director and Collusion do not affect Fraudulent Financial Reporting. In order for the company to have a system to prevent the occurrence of embezzlement, the company created a system of detection, monitoring, and systems that review policies in the field of human resources (HR).

Section 4 begins abruptly, with a series of tables. To increase the flow of ideas and make it easier for readers to understand, it is recommended to add 1-2 paragraphs before the first table in this section, then to explain the results of the descriptive statistical table, and so on. Try to introduce each table, adding a phrase that mentions it.

Has been revised as follows:

  1. Results

Following are the results of Data processed by SPSS 25, 2022 which includes tests, Descriptive Statistics, Collinearity Statistics, Heteroscedasticity Test Results, Hosmer and Lemeshow Test, Coefficient of Determination (Nagelkerke's R Square), Logistics Regression Test Results, Omnibust Test of Models Coefficients (f-test) and WALD Test Results.

  1. The concluding section should be further developed, very short and fast. Thank you for the input that has been provided so that our conclusions develop as follows: Conclusions

This study aims to prove empirically the influence of the stimulus (financial target, financial stability, and external pressure), capability (change in director), collusion (total board of commissioners who have multiple positions), opportunity (ineffective monitoring), rationalization ( auditor switching), and arrogance (frequency of the number of CEO photos in the annual financial statements) against Fraudulent Financial Reporting in banking companies listed on the Indonesia Stock Exchange (IDX) in 2017-2021. Based on the analysis that has been carried out, the conclusions that can be drawn are as follows: the following: External Pressure, Arrogance affects Fraudulent Financial Reporting. However, Financial Target, Financial Stability, Ineffective Monitoring, Auditor Switching, External Pressure, Change in Director and Collusion do not affect Fraudulent Financial Reporting. In order for the company to have a system to prevent the occurrence of embezzlement, the company created a system of detection, monitoring, and systems that review policies in the field of human resources (HR). Therefore, the work unit function who manages human resources or human resources has a very important factor in creating optimization of the implementation of anti-fraud policies. Some policies and procedures Human resources that must be held at least include the process of employee recruitment, education, transparency rotation process, mutation-promotion, sanctioning, eliminating biased policies (not assertive), giving rewards, integrity, remuneration, and performance system.

  1. Suggestions

Future researchers are expected to be able to use other proxies that are more varied in using the Hexagon fraud model. For opportunity variables, institutional ownership, Quality of external and audit Commissioners in the audit commit- tee can be used and for Capability can use Quality of CEO's in a wider population and further research can also add moderating variables to improve research results to be more accurate.

Thank you so much for the time and review that has been given to improve our article. We are very grateful for the guidance, knowledge and extraordinary things to improve our articles so that they become even better.

Best regard

Round 2

Reviewer 4 Report

The authors have improved the paper according to the recommendations. It may be published in this form.